# Blood Transfusion Components Inducing Severe Allergic Reactions: The First Case of Kounis Syndrome Induced by Platelet Transfusion

**DOI:** 10.3390/vaccines11020220

**Published:** 2023-01-19

**Authors:** Christos Gogos, Konstantinos Stamos, Nikolaos Tsanaxidis, Ioannis Styliadis, Ioanna Koniari, Sophia N. Kouni, Cesare de Gregorio, Nicholas G. Kounis

**Affiliations:** 1Department of Cardiology, Papageorgiou General Hospital, Nea Efkarpia, 56403 Thessaloniki, Greece; 2Department of Cardiology, Liverpool Heart and Chest Hospital, Liverpool L14 3PE, UK; 3Speech Therapy Practice, 26221 Patras, Greece; 4Department of Clinical and Experimental Medicine, University of Messina Medical School, 98122 Messina, Italy; 5Department of Internal Medicine, Division of Cardiology, University of Patras Medical School, 26221 Patras, Greece

**Keywords:** allergy, anaphylaxis, blood transfusion, blood components, Kounis syndrome, platelets, transfusion

## Abstract

Kounis syndrome is a multisystem and multidisciplinary disease affecting the circulatory system that can be manifested as spasm and thrombosis. It can occur as allergic, hypersensitivity, anaphylactic, or anaphylactoid reactions associated with the release of inflammatory mediators from mast cells and from other interrelated and interacting inflammatory cells, including macrophages and lymphocytes. A platelet subset with high- and low-affinity IgE surface receptors is also involved in this process. Whereas the heart, and particularly the coronary arteries, constitute the primary targets of inflammatory mediators, the mesenteric, cerebral, and peripheral arteries are also vulnerable. Kounis syndrome is caused by a variety of factors, including drugs, foods, environmental exposure, clinical conditions, stent implantation, and vaccines. We report a unique case of a 60-year-old male with a past medical history of allergy to human albumin, alcoholic cirrhosis, and esophageal varices, who was admitted due to multiple episodes of hematemesis. Due to low hemoglobin levels, he was transfused with 3 units of red blood cells and fresh frozen plasma without any adverse reactions. On the third day of hospitalization, severe thrombocytopenia was observed and transfusion of platelets was initiated. Immediately following platelet infusion, the patient developed chest discomfort, skin signs of severe allergic reaction, and hemodynamic instability. The electrocardiogram revealed ST segment elevation in the inferior leads. Given the strong suspicion of Kounis syndrome/allergic coronary spasm, the patient was treated with anti-allergic treatment only, without any anti-platelet therapy. The clinical status of the patient gradually improved and the electrocardiographic changes reverted to normal. Based on these findings, Kounis hypersensitivity-associated acute coronary syndrome, specifically, type I Kounis syndrome, was diagnosed. Although platelet transfusion can be a life-saving therapy, each blood transfusion carries a substantial risk of adverse reactions. The aims of this report are to expand the existing knowledge of patient responses to blood transfusion and provide information on the incidence of various severe transfusion reactions to all blood components and especially to platelets. To the best of our knowledge, Kounis syndrome induced by platelet transfusionhas never been previously reported. Hypersensitivity to platelet external membrane glycoproteins in an atopic patient seems to be the possible etiology. Despite that Kounis syndrome remains an under-diagnosed clinical entity in everyday practice, it should always be considered in the differential diagnosis of acute coronary syndromes.

## 1. Introduction

Whereas blood transfusion is a life saving therapy for patients with acute blood loss and severe anemia, it carries the risk of a number of adverse reactions, from common transient pyrexias to a potentially fatal ABO mismatch. Despite reducing the risks through advances in patient and staff education, safety checklists, and monitoring, the possibility of transfusion-associated reactions cannot be completely eradicated, except by transfusion avoidance [1]. All blood components can cause severe allergic and anaphylactic reactions, but the most commonly reported reactions are with plasma-rich components such as platelets or fresh frozen plasma. Acute myocardial infarction following ablood transfusion has been rarely reported [2,3,4]. The few reported cases associated with blood transfusions were due to bleeding from peptic ulcers or from gastrointestinal malignancies. The suggested potential mechanisms at that time included: rouleaux formation in the donor red blood cells blocking narrow coronary arteries, rupture of pre-existing atherosclerotic coronary plaque, post-transfusional disseminated intravascular coagulation induced intra-myocardial coronary thrombosis, or formation of coronary thrombus triggered by rapid increases in hematocrit and viscosity. Allergy, hypersensitivity, or anaphylaxis were not associated with these cases. Specific blood components, including platelets, were not incriminated. Indeed, these few reports had been published before the discovery and description of the hypersensitivity and allergy associated with acute myocardial infarction of Kounis syndrome. 

Moreover, in a recent report, the authors described a case of myocardial infarction following a blood transfusion in a patient suffering from anemia and malabsorption syndrome, stating that the clinical experience regarding such cases is “indeed quite limited” [5].

Indeed, in the putative burial site, sarcophagus, and tomb of the Pharaoh Menes of Egypt were found two, partially preserved but almost identical, ebony plates with hieroglyphs of three sketched half circles meaning a hornet wasp, and his death was attributed to a wasp or hornet sting [6]. Menes succumbed unexpectedly while planning to travel to the British Isles in the year 2600 BC [7]. Menes’ death is considered, today, to be the first evidence of anaphylactic death in humans. The first correlation between cardiovascular disorders and anaphylactic reactions was established in experiments performed in the previous century [8]. These experiments were performedin dogs and rabbits but not in guinea pigs because the latter usually died due to asphyxia-induced death during the experiments [9].

The association between cardiovascular symptoms and signs of allergic, hypersensitivity, anaphylactic, or anaphylactoid reactions started to appear, for the first time, in English, German, and Austrian medical literature. Such reactions were mainly attributed to serum sickness and tetanus antitoxins and were characterized as “morphologic cardiac reactions” [10], “acute carditis” [11], or “lesions with basic characteristics of rheumatic carditis” [12].

However, the detailed description of “allergic angina syndrome” as a coronary spasm, which represents a manifestation of endothelial dysfunction or microvascular angina and progresses to allergic acute myocardial infarction, was described in 1991 [13] and later named “Kounis syndrome” [14,15].

This syndrome is caused by inflammatory mediators released during an allergic insult, from activation and degranulation of mast cells and other interacting cells, including T-lymphocytes, macrophages, eosinophils, and platelets [16]. Τhe pathways of triggering the activation and degranulation of mast cells are shown in Table 1.

Histamine, tryptase, and arachidonic acid products, in combination with chymase acting as a converting enzyme, can promote acute ischemic events via coronary spasm, atheromatous plaque erosion/rupture, and platelet activation in the Kounis syndrome cascade. A platelet subset of 20% with high- and low-affinity IgE surface receptors is also involved in this process [20,21].

Potential inciting causes of Kounis syndrome include drugs, vaccines, metals, foods, environmental exposure, and clinical conditions. Kounis syndrome can affect not only the coronary arteries, but also the mesenteric, cerebral, and peripheral arteries, with an incidence ranging from 1.1% to 3.4% in patients who suffer an allergic, hypersensitivity, anaphylactic, or anaphylactoid insult [22]. Kounis syndrome was first thought to be a rare condition but appears rather to be an under-diagnosed disease. Three types of this syndrome have been described to date, as summarized in Figure 1.

The aims, therefore, of this report are to expand the existing knowledge of patient responses to new causes able to induce allergic reactions and Kounis syndrome, including blood transfusion, and provide information about the incidence of various severe transfusion reactions to all blood components and especially to platelets. A unique case of this syndrome induced during platelet transfusion is described below.

## 2. Case Presentation

A 60-year-old male with a past medical history of alcoholic cirrhosis was referred to the hospital’s emergency department with altered mental status and multiple episodes of hematemesis and melena. The patient’s medical history included multiple admissions for alcohol-related problems, including two previous episodes of bleeding from esophageal varices. In addition, in a recent hospitalization, he suffered a severe allergic reaction following intravenous administration of human albumin. He was on treatment with carvedilol 6.25 mg t.i.d., rifaximin 400 mg t.i.d., spironolactone 100 mg s.i.d., and norfloxacin 400 mg s.i.d. On admission, the patient was critically ill with hypotension (blood pressure 80/40 mmHg), cold extremities, hypothermia (34.8 °C), tachycardia (heart rate 135 beats/min), tachypnea (respiratory rate 35/min), and his Glasgow Coma Scale was 12/15. The oxygen saturation was 99% on room air. His physical examination was unremarkable. Initial laboratory results demonstrated leukocytosis (WBC- 22.72 × 103/µL) with neutrophilia (85.5%), low hemoglobin (6.9 g/dL), thrombocytopenia (platelet count 20 × 109/L), and increase in D-dimer (11.510 ng/mL [normal range < 0.05 ng/mL]) levels. Moreover, total bilirubin was 6.3 mg/dL (normal range 0.2–1.3 mg/dL) and serum albumin was 1.9 g/dL (normal range > 3.5 g/dL). Liver enzymes, creatinine, and potassium levels were normal. Patient was gradually stabilized with intravenous fluids and transfusion of 2 units of red blood cells without any adverse events. Additionally, two units of fresh frozen plasma (FFP), intravenous esomeprazole sodium (40 mg), ceftriaxone (2 g), and somatostatin (3 mg) were administered. An urgent upper gastrointestinal endoscopy demonstrated the absence of active variceal bleeding. On the second day of hospitalization, 1 unit of fresh frozen plasma and red blood cells were also given without any adverse reactions. On the third day, despite significant clinical improvement, a severe decrease inplatelet levels(platelet count 20 × 109/L) was observed and platelet transfusion was considered mandatory. Soon after starting platelet transfusion, the patient developed chest discomfort followed by diffuse burning skin sensation, shortness of breath, facial edema, wheezing, widespread skin rash, and throat swelling, indicative of severe anaphylactic reaction with laryngeal edema. The patient’s blood pressure was undetectable. The emergency electrocardiogram revealed ST segment elevation in inferior leads, as well as a ST segment depression in anterior leads with first-degree atrioventricular block (Figure 2).

Immediately, intramuscular epinephrine 0.5 mg (0.5 mL, 1:1000) wasadministered. Additionally, hydrocortisone and dimetindene were intravenously initiated, while the anti-platelet therapy was discontinued. Following administration of anti-allergic treatment, a new electrocardiogram showed complete resolution of ST-segment elevation (Figure 3) and the patient’s clinical status gradually improved.

A transthoracic echocardiogram revealed a normal ejection fraction, with no regional wall motion abnormalities. High-sensitivity cardiac troponin I (hs-TnI) levels at 6 and 24 h were normal. Based on these findings, Kounis hypersensitivity-associated acute coronary syndrome, specifically, type I Kounis syndrome, was suspected (coronary artery spasm o rmyocardial infarction with non-obstructive coronary arteries (MINOCA). Additional allergy tests, such the basophil activation test, radioallergosorbent test, enzyme-linked immunosorbent assay, and fluoroenzyme immunoassay, were not available. Coronary angiography was not performed as the patient and relatives refused consent. The patient declined further hospitalization and was self-discharged home.

## 3. Discussion

Allergens are foreign proteins or glycoproteins that constitute the target of IgE antibody responses in humans. On the other hand, platelets are small, colorless, plate-like cells from which their name is derived. Their peripheral zone is rich in glycoproteins, such as GPIIb/IIIa, GPIb/IX/V, and GPVI, which are necessary for platelet adhesion, activation, and aggregation. Such glycoproteins are often important integral membrane proteins and play an important role in cell-to-cell interactions. Recent experimental studies have demonstrated that blocking of the GP IIb/IIIa glycoprotein receptor can significantly inhibit platelet–eosinophil aggregation as well as platelet and eosinophil activation, thereby attenuating airway hyper-responsiveness and airway inflammation in an eosinophilic asthma model. The results of this study suggested that the GP IIb/IIIa receptor can be used as a novel therapeutic target for asthma [23]. Overall, understanding allergic diseases requires essential knowledge regarding the source, particles, and routes of exposure, as well as the properties of the individual allergens [24]. The described patient had a history of allergic reaction to albumin; therefore, he had an atopic diathesis to any allergen and was prone to allergic and anaphylactic reactions and Kounis syndrome development.

### 3.1. Types and Incidence of Blood Transfusion Reactions

Blood transfusion is one of the most common life-saving procedures performed in hospitals today [25]. Some authors have characterized the administration of blood or its components from one person to another as a form of tissue transplantation [26]. Acute transfusion reactions occur within 24 h of transfusion, including [27,28]:Allergic transfusion non-hemolytic reactionsAltered oxygen affinityAcute hemolytic transfusion reactionsDelayed hemolytic transfusion reactionsDelayed serologic transfusion reactionsHypotensive transfusion reactions Infections Post-transfusion purpuraTransfusion-associated graft vs. host diseaseTransfusion-transmitted infectionTransfusion-associated dyspneaTransfusion-associated circulatory overloadTransfusion-related acute lung injuryMetabolic reactions

The common form of acute transfusion allergic reactions can manifest as urticaria, pruritus, erythematous rash, angioedema, bronchospasm, and hypotension, which can lead to anaphylactic shock [29]. The incidence of anaphylactic reactions to blood products is 1 per 20,000–50,000 [30]. The incidence of allergic transfusion reactions is 0.3% to 6% for platelet transfusion and 1% to 3% for plasma transfusion [31]. There is a difference between children and adults in the frequency and distribution of transfusion-related adverse reactions. The incidence of transfusion reactions is higher in pediatric patients than in adults, especially for red blood cell and platelet transfusions [31].

### 3.2. Etiology of Platelet Transfusion Induced Allergy Reactions

Platelets are derived from megakaryocytes and constitute small anucleate cellular fragments. One trillion platelets circulate in human blood [32]. Platelets are considered immune cells participating in various immune-related processes and play critical roles in hemostasis and thrombosis. Experimental evidence has demonstrated that platelets can release a variety of mediators, including histamine [33], serotonin [34], thromboxane A_2_ [35], platelet-activating factor [36], IL-33 [37,38], and a variety of growth factors [39,40]. Moreover, platelets can exacerbate airway inflammation and constitute a contributor in the development of allergic asthma [41]. There are several possible reasons for the association between platelets and allergic and anaphylactic reactions during blood transfusion, such as: (1) The existence of IgE and IgG antibodies in the recipient against plasma proteins in the transfused blood component [41]. (2) The transfusion of cytokines, chemokines, and histamine generated in the platelet product during preparation and storage [41]. (3) Antibodies against antigens on granulocytes and HLA Class I and II antibodies may induce transfusion-related acute lung injury (TRALI), usually in female patients, resulting in dangerous, sometimes fatal transfusion reactions [41]. (4) Platelet-specific alloantibodies may exist in blood plasma, especially from female donors. Transient episodes of immune thrombocytopenia, also called passive alloimmune thrombocytopenia, have been observed in patients with normal platelet counts receiving fresh frozen plasma with platelet alloantibodies. Since the patients who need platelet transfusion are usually thrombocytopenic, as in our patient, such reactions will probably not be noted as a consequence of platelet transfusions [41]. (5) A similar situation exists in the case of post-transfusion purpura (PTP) observed in (mostly) female patients after transfusion of whole blood or red cell concentrates in the past [41]. (6) Contaminated T-lymphocytes in platelet concentrates may also cause transfusion-associated graft-versus-host disease (TA-GvHD) in a similar manner as transfusion of red blood cells [41]. (7) The allergic characteristics and history of atopy of the patient receiving the transfusion and the cellular and plasma components of the transfusion products can contribute to the development of allergic reactions [41,42]. Indeed, our patient had suffered severe allergic reaction in the past following intravenous administration of human albumin, thus confirming his atopic diathesis for the development of Kounis syndrome. (8) The existence of specific medical conditions, such as IgA or haptoglobin deficiency, can make patients more susceptible to anaphylaxis [43]. (9) An equivalent to anti-IgA-induced anaphylactic transfusion reactions has been also observed in haptoglobin-deficient individuals. Haptoglobin deficiency was identified in 367 (1.6%) patients with a history of sudden onset of anaphylactic transfusion reactions [44].

Kounis syndrome, therefore, represents not a single-organ arterial disorder but a multisystem, multidisciplinary, and multi-etiological disorder. Some strange, rare causes of Kounis syndrome are described in Table 2.

## 4. Conclusions

Although platelet transfusion can be a life-saving therapy, every transfusion carries a substantial risk of adverse reactions. However, mild platelet-induced allergic reactions without hypotension and in the absence of skin manifestations [64] may be missed in patients who have already been markedly thrombocytopenic before platelet transfusion.

To the best of our knowledge, Kounis syndrome induced by platelet infusion has never been reported. The described patient had a past history of allergy to human albumin; therefore, hypersensitivity to platelet external membrane glycoproteins in an atopic patient seems to be the possible etiology. Despite the fact that Kounis syndrome still remains an under-diagnosed clinical entity in everyday practice, it should always be considered in the differential diagnosis of acute coronary syndromes. 

## Figures and Tables

**Figure 1 vaccines-11-00220-f001:**
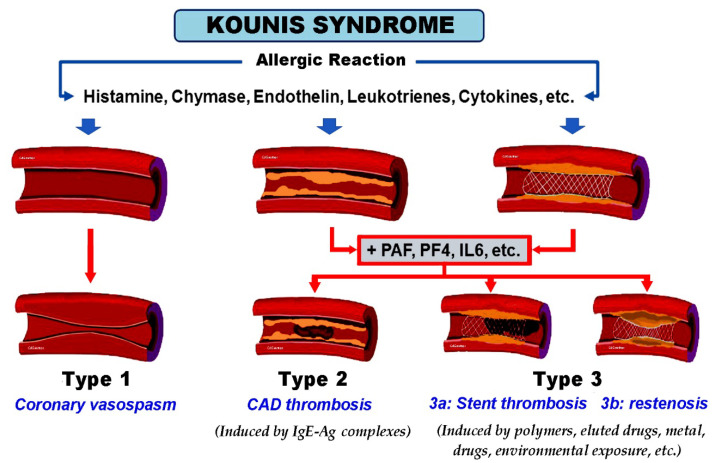
Kounis syndrome. Current classification in 3 subtypes based on allergic pathogenic mechanisms and ensuing effects. Type 1 usually occurs in normal or near-normal coronary arteries, largely due to histamine degranulation from mast cells, catecholamine storm, endothelin release, and other cytokines (MINOCA syndrome). Type 2 is usually seen in patients with quiescent pre-existing coronary artery disease (CAD) and may result in acute coronary syndrome/myocardial infarction. Type 3 affects patients with coronary artery stenting, with subsequent acute thrombosis or restenosis. Ag, antigen; IgE, immunoglobulin E; IL6, interleukin 6; MINOCA, myocardial infarction with non-obstructive coronary arteries; PAF, platelet activation factor; PF4, platelet factor 4.

**Figure 2 vaccines-11-00220-f002:**
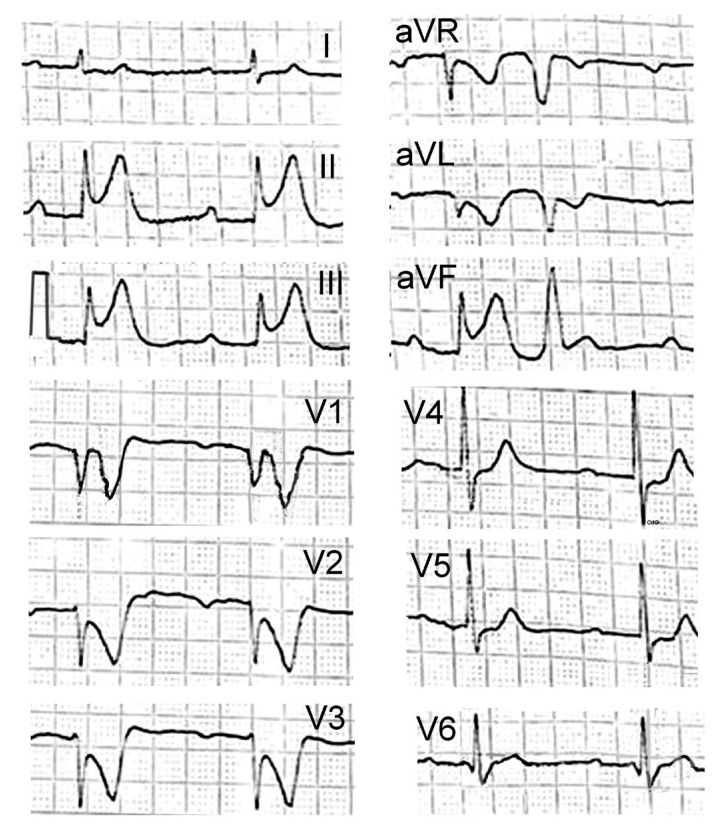
The emergency electrocardiogram, performed under anaphylactic shock conditions, showing ST segment elevation in the inferior leads, as well as ST segment depression in the anterior leads with first-degree atrioventricular block.

**Figure 3 vaccines-11-00220-f003:**
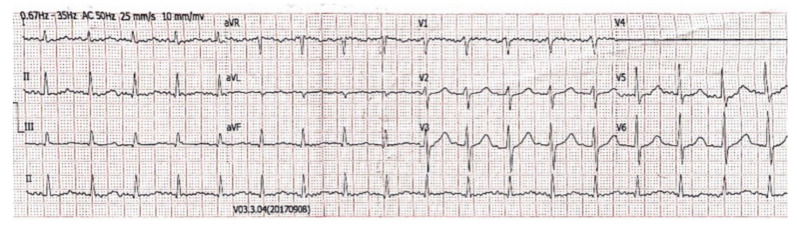
Following administration of anti-allergic treatment, a new electrocardiogram showed sinus tachycardia with complete resolution of ST-segment elevation and first-degree atrioventricular block.

**Table 1 vaccines-11-00220-t001:** Pathways of activation and degranulation of mast cells.

1. Antigens crosslinking antigen-specific IgE bound to high affinity FcεR1 (Fragment Crystallizable epsilon Region 1) receptors [17].
2. Non-IgE-mediated mast cell degranulation via activation of complement C1q, C3a C4, C5a, and Factor B, which are called anaphylatoxins. This pathway involves IL-5 and tryptase and is common in patients who develop renal failure or fatal cerebral events [17].
3. Low-affinity mas-related G protein-coupled receptor X2 (MRGPRX2) may activate mast cells via non-FcεR1 receptors [18].
4. Neuropeptides, including the corticotropin-releasing hormone, neurotensin, and substance P via high-affinity receptors during psychological stress conditions [19].

**Table 2 vaccines-11-00220-t002:** Strange, rare causes of Kounis syndrome.

**Administrations**
Acetaminophen Infusion [45]
Antivenom administration [46]
Azithromycin intravenous administration [47]
Clopidogrel administration (drug that treats cardiac ischemia!) [48]
Contrast intravenous media [49]
Coronary angiogram (iodinated contrast medium) [50]
Gadolinium (diagnostic drug for cardiac diseases that contains polyethylene glycol (PEG), an excipient of COVID-19 vaccines) [51]
Laxative administration (contains polyethylene glycol (PEG), an excipient of COVID-19 vaccines) [52]
**Bites and stings**
Cobra bite (cardiogenic shock) [53] Leech bite therapy [54]
Octopus bite [55]
Pigeon tick *Argas reflexus* bite (near-fatal) [56]
Bee stings [57] Scorpion sting [58]
**Consumptions**
Aspirin consumption [59]
Blue crab consumption [60]
Salad consumption [61]
**Latex**
Latex contact [62]
Latex intraoperative [63]

## Data Availability

Not applicable.

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
