# Peer review of "Blood Transfusion Components Inducing Severe Allergic Reactions: The First Case of Kounis Syndrome Induced by Platelet Transfusion"

_vaccines, 2023, doi:10.3390/vaccines11020220_

Round 1

Reviewer 1 Report

Originality:  This is a very paper, providing a maximum of information about a case study, and constitutes a very important contribution to the literature.  I think it would be a more clear case report if the following parts were revised and supplemented. These will be discussed below relative to the information of the manuscript.

Specific comments:

Title: The title of this manuscript is very short. Perhaps a  that include the type of this paper is more appropriate version for clarity, interes and ease of read.

Abstract: It is hard to get the detail in an abstract when the word count is limited and this is often the hardest part of a paper to write. However, I do feel that it would be beneficial to explain the aim and conclusions what specifically you are looking at in relation to outcomes in this case report. This needs to be made clearer throughout the paper

Keywords: Please use recognised MeSH terms as this will assist others when they are searching for information on your research topic. The following website will provide these (simply start typing in a keyword and see if it exists or find an alternative if it does not): https://www.ncbi.nlm.nih.gov/mesh

The introduction is weak and very short. An introduction should announce your topic, provide context and a rationale for your work, while catching the reader´s interest and attention. The above has not been given in the introduction that I have read.

Also, please describe the objective in this section.

Methodologically Sound:  As a case study report it is rather hard to go wrong methodologically, and the paper conforms to the standard.

Follows Appropriate Ethical Guidelines: Whilst there is no obvious declaration of ethical approval. Please include the date and code register number of ethics committee  it would appear to be a report of actions taken as part of normal clinical practice (as a case study report), and thus is acceptable. 

Has results which are clearly presented and support the conclusions: Again, it conforms to the usual format for the presentation of a case study, although the content is very long.  It is, however, appropriate enough, and does report a rare case likely to be of interest to a healthcare audience.  

Overall Scientific Quality:  As a case study report it have scientific depth, but effectIvely is intended only to report the occurrence of a typical case and to highlight the importance of correct diagnosis, and on these grounds merits attention. 

Presentation, Organization, Clarity:  I think you have some good information. 

Correctly References Previous Relevant Work:  It appears to reference prior work succinctly and accurately. 

Importance/Interest: Although marked by its brevity, the content is of interest, particularly to clinicians and hematologistswho examine syndrome a great deal of the time, and nursers  who may need to be aware of the variant forms of this illness.  

Author Response

REVIEWER 1

Specific comments:

All corrections are depicted in red color in the revised version

Title: The title of this manuscript is very short. Perhaps a title that include the type of this paper is more appropriate version for clarity, interests and ease of read.

Thank you. we have expanded the title as: Severe Allergic Reactions Induced by blood Transfusions: The first case of Kounis syndrome induced  by  platelet  transfusion

Abstract: It is hard to get the detail in an abstract when the word count is limited and this is often the hardest part of a paper to write. However, I do feel that it would be beneficial to explain the aim and conclusions what specifically you are looking at in relation to outcomes in this case report. This needs to be made clearer throughout the paper

Thank you we have added the following: The aims of this report are to expand the existing knowledge on patients’ responses to blood transfusion and to provide information on the incidence of various severe transfusion reactions to all blood products and especially to platelets

Keywords: Please use recognised MeSH terms as this will assist others when they are searching for information on your research topic. The following website will provide these (simply start typing in a keyword and see if it exists or find an alternative if it does not): https://www.ncbi.nlm.nih.gov/mesh

Thank you,  we have added some key words according to MeSH terms

The introduction is weak and very short. An introduction should announce your topic, provide context and a rationale for your work, while catching the reader´s interest and attention. The above has not been given in the introduction that I have read. Also, please describe the objective in this section.

Thank you,  we have expanded the introduction as advised,  according the word limits of VACCINES for case reports paper M/S.

 Follows Appropriate Ethical Guidelines: Whilst there is no obvious declaration of ethical approval. Please include the date and code register number of ethics committee  it would appear to be a report of actions taken as part of normal clinical practice (as a case study report), and thus is acceptable. 

Thank you,  we have added it

Reviewer 2 Report

In this manuscript, the authors report the first case of Kounis syndrome induced by platelet infusion in a 60-year-old male. This is an interesting case report and the results could be valuable for other researchers and physicians in this field. Some comments were suggested as follows.

1. The present abstract is not concise; Please try to refine it and highlight the conclusions.

2. The introduction should be intensively improved since it routinely introduces some historical development of Kounis syndrome without real scientific depth. The authors even use three paragraphs (Line 50-69) to tell the history of Kounis syndrome. 

3. Line 79: “A platelet subset of 20% with high- and low-affinity IgE surface receptors is also involved in this process”. Please provide the supporting reference.

4. Case presentation: The complete timeline from the onset of symptoms in this patient to discharge should be provided.

5. There are many relevant reasons to associate platelets with allergic and anaphylactic reactions during blood transfusion. For the case in this report, what is the possible reason for the Kounis syndrome induced by platelet infusion?

Author Response

REVIEWER 2

All corrections are depicted in red color in the revised version

In this manuscript, the authors report the first case of Kounis syndrome induced by platelet infusion in a 60-year-old male. This is an interesting case report and the results could be valuable for other researchers and physicians in this field. Some comments were suggested as follows.

  1. The present abstract is not concise; Please try to refine it and highlight the conclusions.

Thank you,  we have added the aims and  conclusion

  1. The introduction should be intensively improved since it routinely introduces some historical development of Kounis syndrome without real scientific depth. The authors even use three paragraphs (Line 50-69) to tell the history of Kounis syndrome. 

Thank you, we have tried to revise the introduction as advised

  1. Line 79: “A platelet subset of 20% with high- and low-affinity IgE surface receptors is also involved in this process”. Please provide the supporting reference

Thank you,the reference has been added.

  1. 4. Case presentation: The complete timeline from the onset of symptoms in this patient to discharge should be provided.

Thank you, in limit number of words for case reports of VACCINES, we have done our best

  1. There are many relevant reasons to associate platelets with allergic and anaphylactic reactions during blood transfusion. For the case in this report, what is the possible reason for the Kounis syndrome induced by platelet infusion?

Thank you,  allergy to glucoproteins of platelet external membrane in an atopic patient seems to be the possible etiology.

Round 2

Reviewer 1 Report

I am happy with the paper as it stands. Congratulations.